# Response of Aphid Parasitoids to Volatile Organic Compounds from Undamaged and Infested *Brassica oleracea* with *Myzus persicae*

**DOI:** 10.3390/molecules27051522

**Published:** 2022-02-24

**Authors:** Qasim Ahmed, Manjree Agarwal, Ruaa Alobaidi, Haochuan Zhang, Yonglin Ren

**Affiliations:** 1Department of Plant Protection, College of Agricultural Engineering Sciences, Al-Jadriya Campus, University of Baghdad, Baghdad 10071, Iraq; qasim.h@coagri.uobaghdad.edu.iq; 2Department of Agricultural Sciences, College of Science, Health, Engineering and Education, Murdoch University, South Street, Murdoch, WA 6150, Australia; m.agarwal@murdoch.edu.au (M.A.); 32985118@student.murdoch.edu.au (H.Z.); 3Department of Clinical Laboratory Sciences, College of Pharmcy, Al-Qadisyia Campus, Al-Mustansiriyah University, Baghdad 10052, Iraq; ruaa_abdulsattar@uomustansiriyah.edu.iq

**Keywords:** green peach aphids, VOCs, parasitoids, *Aphidus colemani*, *Aphelinus abdominalis*, cabbage

## Abstract

Headspace solid microextraction (HS-SPME) and GC-MS were used to investigate volatile organic compounds (VOCs) from cabbage plants infested and uninfested with green peach aphid *Myzus persicae*. The HS-SPME combined with GC-MS analysis of the volatiles described the differences between the infested and uninfested cabbage. Overall, 28 compounds were detected in infested and uninfested cabbage. Some VOCs released from infested cabbage were greater than uninfested plants and increased the quantity of the composition from infested plants. According to the peak area from the GC-MS analysis, the VOCs from infested cabbage consisted of propane, 2-methoxy, alpha- and beta pinene, myrcene, 1-hexanone, 5-methyl-1-phenyl-, limonene, decane, gamma-terpinen and heptane, 2,4,4-trimethyl. All these volatiles were higher in the infested cabbage compared with their peak area in the uninfested cabbage. The results of the study using a Y-shape olfactometer revealed that the VOCs produced by infested cabbage attracted *Myzus persicae* substantially more than uninfested plants or clean air. The percentage of aphid choice was 80% in favor of infested cabbage; 7% were attracted to the clean air choice and uninfested plants. A total of aphids 7% were attracted to clean air. Comparing between infested and uninfested cabbage plants, the aphid was attracted to 63% of the infested cabbage, versus 57% of the uninfested cabbage. The preferences of *Aphidus colemani* and *Aphelinus abdominalis* to the infested or uninfested plants with *M. persicae* and compared with clean air indicated that parasitoids could discriminate the infested cabbage. Both parasitoids significantly responded to the plant odor and were attracted to 86.6% of the infested cabbage plants.

## 1. Introduction

*Myzus persicae* (Hemiptera: Aphididae) has a universal distribution, including Australia, and is considered a serious pest that has caused damage to hundreds of agricultural crops in more than 66 families [1,2]. The aphid mainly exists in young plant tissues, causing reduced leaf size, delayed growth of the plant and reduced yield [3]. *M. persicae* is considered a common pest insect of cruciferous crops, and sucks plant sap, leading to yellowing and curling of plant leaves. Additionally, the excretion of honeydew by aphids affects plant photosynthesis and encourages fungal growth [4]. Cabbage plants are commonly attacked by different species of aphids, such as turnip aphid *Lipaphis erysimi*, cabbage aphid *Brevicoryne brassicae* and green peach aphid *M. persicae*, which economically damage these crops [5].

Chemical insecticides play a significant role in controlling insects on crop plants. Insecticides have been extensively used in horticultural systems; however, they can cause the appearance of secondary pests instead of primary pests, pesticide resistance, contamination of environment and affect non-target organisms [6,7]. Therefore, it is necessary to find alternative methods for pest management. In biological control, aphid parasitoids from families such as Braconidae and Aphelinidae are important and can cause a high percentage of mortality on aphids [8,9]. Natural enemies of aphids can reduce the rate of population increase, and the use of wasp parasitoids in biological control of aphids has been successful [10].

Plants VOCs play an important role in plant–insect interactions by influencing insect communication and plant defense [11]. When sucking insect pests such as the green peach aphid feed on the plant, one response from the plant is to release odors in the form of VOCs. The VOCs have an important role in plant–insect interactions because they can be used by parasitoids to locate their host [12].

Cabbage plants attacked by aphids may emit volatile compounds that attract parasitoid wasps or predators [13,14]. Previous studies have concluded that natural enemies can identify the VOCs released from the infested plants; the response of parasitoids and predators were confirmed and this provided an explanation how natural enemies were attracted by the host plant using the olfactory scale [15,16].

*Aphidius colemani* (Hymenoptera, Braconidae) and *Aphelinus abdominalis* (Hymenoptera, Aphelinidae) are endoparasitoids of many species of aphids and both attack *M. perisecae* [17]. The VOCs released from infested *Brassica* plants by aphids can bring with lure parasitoids, which showed the family of Brassicaceae possess chemical defense [18]. When the aphids feed on the plant leaves, the plant produces blends of volatiles as a response to the infestation by aphids, releases volatile compounds in different quantities and qualities from damaged Brassica plants, and these differences in the VOCs can attract other pests and natural enemies [17]. *A. colemani* and *A. abdominalis* are parasitic wasps specific to green peach aphids, whose females use VOC signals to detect and locate aphids feeding on host plants and lay their eggs into aphids [19,20]. Additionally, honeydew excreted by aphids on plants could lead to the release of semiochemicals or VOCs attracting and guiding parasitoids to the aphid [21,22].

In Y-tube olfactometer tests, Reed [8] reported no attraction of the parasitoid *Diaeretiella rapae* to the cabbage leaves. However, the choice of wasps to infest cabbage plants by *B. Brassicae* was more significant than other plants infested by different species of aphid, such as Russian wheat aphid *Diuraphis noxia*. These results indicate that the cabbage plant VOCs are more important than other plants in attracting the parasitoid to the aphid location [8]. The heavy population of *M. persicae* on the plant can accumulate wasps, while the uninfested plant sees few parasitoids come to the plant because wasps fail to locate the uninfested plant [23,24].

The identification of VOCs can be a signal for aphids and their parasitoids’ receivers, and it is necessary to develop methods to analyze VOCs as diagnostic indicators that involve aphid management. Therefore, this study aims to determine the VOCs released from *M. persicae*-infested and uninfested cabbage plants to elucidate the responses of *M. persicae* and their parasitoids (*A. colemani* and *A. abdominalis*) to aphid-infested and uninfested cabbage plants in the Y-tube olfactometer. Understanding the treatments influencing the attraction of the parasitoids may provide fundamental data for controlling green peach aphids and generating new methods for aphid biological control.

## 2. Results

### 2.1. VOCs Released from M. persicae Infested and Uninfested Plant

Analysis of the volatiles of cabbage induced by *M. persicae* for the infested and uninfested plant treatments shows significant differences. Several compounds were present in all samples that were trapped by SPME and identified by GC-MS. Plants damaged by *M. persicae* can change in plant odor emission, and the volatiles of samples were significantly higher than uninfested plants. The volume and the variety of VOCs released from infested cabbage were greater than the uninfested plant in some compounds, and the qualitative differences in the composition of the odor from these treated plants consisted of propane, 2-methoxy that was released from uninfested cabbage, which was greater than the VOCs released from infested cabbage, with an average peak area in the uninfested plant of 23.10 compared with the peak area in the infested plant of 7.84. Meanwhile, alpha- and beta pinene were much higher in the infested than uninfested plants (Table 1). There was a significantly larger quantity of (*E*)-3-hexen-1-ol (*p*-value 0.223), beta-pinene (*p*-value 0.930) and decane (*p*-value 0.020) released from the infested plant but not detected in uninfested cabbage plants. Moreover, the peak area for the following volatile compounds, which were detected from infested cabbage, were higher in the infested cabbage compared with their peak area in the uninfested cabbage: myrcene, 1-hexanone, 5-methyl-1-phenyl-, limonene, decane, gamma-terpinen and heptane, 2,4,4-trimethyl. However, some of the volatile compounds from uninfested cabbage were released in a high amount based on peak area detected by GC-MS as compared with the infested plant. These compounds were eucalyptol, cyclohexasiloxane, 3,4-dihydroxyphenylglycol, 1,5-pentanediamine, octamethyl and decamethyl. VOCs lead to odor differences between aphid infested plants and uninfested plants. Figure 1 shows the heat map that graphically displays results by hierarchical clustering of the volatile compounds from the infested and uninfested cabbage. This work was conducted to find the closeness of individual compounds released from both samples (uninfested and infested plants with *M. persicae*). Distances between samples and assays were calculated for hierarchical clustering based on Pearson’s Correlation Distance. Each volatile compound has a peak area detected by GC-MS, presented by the color scale that illustrates the differences between the replicates of the infested and uninfested cabbage. The heat map indicated that the detected compounds and the difference between uninfested and infested cabbage plant with the scale of color and each color corresponds to one detected VOC. The value of the compound is represented by red, orange and dark blue for the maximum (2), average (0) and minimum (–2) (Figure 1). In addition, principal component analysis (PCA) was performed and the PCA score plot (Figure 2) shows the separation of the two samples (uninfested and infested plants with *M. persicae*) into two different groups based on their profile of volatile organic compound using the significant difference (*p* < 0.05), relationship between the VOCs within infested and uninfested as shown in Figure 2.

### 2.2. Effect of VOCs on Attractive Parasitoid

Results of the laboratory experiments using Y-tube olfactometer bioassays showed the response of the aphids *M. persicae* (*n* = 30 for each replicate) and their parasitoids *A. colemani* and *A. abdominalis* (*n* = 15 for each replicate and each parasitoid) to the uninfested and infested cabbage plants by 30 individual aphids and 15 individuals per replicate of parasitoid.

These results indicated that green peach aphids in cabbage were significantly (Chi-square (χ^2^) = 18.61, df = 1 and *p* < 0.0005) more attracted to the VOCs released from infested plant (80%) rather than clean air (7%). Results showed that *M. persicae* were significant different in the preference for cabbage plants, with more attraction to the uninfested plants than clean air. The percentage of attracted aphids was 75.56% versus 3% (χ^2^ = 20.16, df = 1 and *p* < 0.0005). While the results indicated that the aphids were significantly more attracted to the infested cabbage compared with the uninfested plant, the percentage of aphid numbers attracted towards infested cabbage plants was 63%, versus 26.67% attracted to uninfested cabbage plants (χ^2^ = 4.48, df = 1 and *p* < 0.034) (Figure 3).

For the parasitoid experiments, the attraction of parasitoids *A. colemani* and *A. abdominalis* to volatiles released by plants, where they were given a choice between uninfested and infested plants, was analyzed. Both *A. colemani* and *A. abdominalis* were significantly more attracted to volatiles from plants infested with green peach aphids compared with clean air (Figure 4). The frequency of parasitoid attraction was 93.33% and 100% towards the infested cabbage plant versus 7% and 20%, respectively, towards the clean air for both parasitoids *A. colemani* and *A. abdominalis* (χ^2^ = 11.26, df = 1 and *p* = 0.001 for *A. colemani* and χ^2^ = 4.57, df = 1 and *p* = 0.033 for *A. abdominalis*). The statistical analysis showed that both parasitoids were significantly attracted to the infested plant. However, there was no difference between attracted wasps for the odors released from an uninfested plant and clean air, and there were no responses for both parasitoids *A. colemani* and *A. abdominalis* to the healthy plant odor versus clean air (both parasitoids showed no significant response to the treatment). By percentage, 4.44% of *A. colemani* wasp and 7% of *A. abdominalis* were attracted to volatiles released from uninfested plants, versus 7% for both parasitoids headed for clean air treatment, while the percentage of no responses of parasitoids was 88.86% and 86.66% for *A. colemani* and *A. abdominalis*, respectively (χ^2^ = 19.20, df = 2 and *p* = 0.001 for *A. colemani* and χ^2^ = 19.20, df = 2 and *p* = 0.001 for *A. abdominalis*). When given a choice between uninfested and infested cabbage plants, *A. colemani* and *A. abdominalis* parasitoids were significantly more attracted to volatiles released from infested plant rather than attracted towards uninfested cabbage plants. By percentage, 86.67% of the *A. colemani* and 100% of the *A. abdominalis* responded to infested cabbage compared to 9% of the *A. colemani* and 0% of the *A. abdominalis* being attracted to uninfested plants (χ^2^ = 10.28, df = 1 and *p* = 0.001 for *A. colemani* and χ^2^ = 12.25, df = 1 and *p* = 0.0005 for *A. abdominalis*).

## 3. Discussion

The VOCs that released from infested cabbage plants by *M. persicae* showed many compounds comparing with uninfested plants and reported by previous studies [25,26,27]. In the current study, volatile compound profiles of uninfested and infested cabbage plants with *M. persicae* were compared to show the differences between treated plants and used as identification tools for the infestation. Taveira et al. [27] reported that a comparison of volatile compounds identified from uninfested and aphid-infested plants from several *Brassica* plants. The damage of cruciferous plants caused by aphids can emit many volatile compounds such as glucosinolate metabolites, phenolics and terpenoids [28,29]. However, our results showed the *M. persicae* preferred damaged *Brassica* plants because the infested plant released different VOCs, such as alpha- and beta pinene, €-3-hexen-1-ol, myrcene, 1-hexanone, 5-methyl-1-phenyl, limonene, decane, gamma-terpinen and heptane, 2,4,4-trimethyl. This finding is consistent with [17], who reported that alpha- and beta pinene and limonene could increase in *Brassica* plants infested by aphids. Some VOCs disappeared from uninfested plants, such as 3-hexen-1-ol-(E) and beta-pinene [17,30]. The increase in (E)-3-hexen-1-ol, beta-pinene and decane in infested plants could be expected because these compounds are well known as green leaf volatiles and are involved in the attraction of natural enemies such as parasitoids and predators [22,31]. The VOCs can be released by an intact and uninfested Brassicaceous plant in large amounts [32]. These compounds were found in the headspace of infested cabbage plants and can be involved in attracting beneficial insects as a response to the aphid infestation [22,31,33]. Thus, the selection of SPME in the extraction of volatile compounds from uninfested and infested cabbage plants with *M. persicae* was based on the peak areas of all compounds identified in the treatments.

The results of Y-tube olfactometer bioassays confirmed the results of aphids *M. persicae* and the parasitoids *A. colemani* and *A. abdominalis* were influenced and attracted to volatiles produced by Cruciferous plants. These wasps significantly preferred, and were attracted to, volatiles from aphid-infested plants over uninfested plants. The use of the Y-tube olfactometer to test the response of aphid *M. persicae* to the host plant, *B. oleracea* var.* capitata*, indicated that *M. persicae* was influenced by the volatiles released from *B. oleracea* var. *capitata* and were significantly attracted to both uninfested and infested plants when compared with the clean air choice.

Aphids can find their host visually and chemically, by chemical, color, size and the shape of the host, and this may be a useful guide to attracting aphids. This result confirms past studies [20,34,35] that show aphids find their host plants by plant odor as well as visual cues. Moreover, the attraction of aphids to the plant volatiles using olfactometer has been reported in experiments testing plant odor against aphids and their host-finding ability [34,35,36]. Our results showed that aphids tended to be attracted to both damaged and undamaged plants. Our observation is that plant compounds can explain the variance in attraction by aphids and also that plant volatile compounds can increase in response to feeding [37,38]. The population of natural enemies can be increased when adding organic fertilizer [39]. Based on VOCs from cabbage, *M. persicae* was attracted to seven different cabbage varieties in diverse ways. Additionally, the wingless *M. persicae* was considerably attracted to Qingan 80 cabbage cultivar in Y-tube olfactorometer bioassays as compared to Yuanbao cabbage cultivar [40,41].

The results from the olfactometer studies demonstrated that parasitoids respond to the plant volatiles and that *A. colemani* and *A. abdominalis* respond to the odor released from infested plants. Both tested parasitoids are significantly responsive to plant volatiles when compared with a clean air treatment. This finding is consistent with [42]. The preference of *A. colemani* and *A. abdominalis* showed no response of parasitoid attraction to clean air and uninfested cabbage, while a statistically significant non-response was noted in the parasitoids. van Emden et al. [43] explain that the attraction of parasitoids can be significantly higher to the infested plant and attack aphids feeding on the same plant as the origin of the mummy offered. The parasitoids *A. colemani* and *A. abdominalis* showed their responses to the infested *B. oleracea*, preferring aphid-induced volatiles. Both parasitoids have significant responses to infested plants with aphids. The results are consistent with [44] who showed parasitoid *A. colemani* could be attracted to volatiles released from *Brassica juncea* and preferred plants damaged by green peach aphids rather than plants damaged by *M. persicae* and *Plutella xylostella* caterpillars.

## 4. Materials and Methods

### 4.1. Experimental Plants

Cabbage (*Brassica oleracea* L. var. *capitata*) seeds were sown in a 90 mm square pot filled with potting soil mixture (Richgro Regular Potting Mix, NSW, Australia) and grown under greenhouse conditions at 23–25 °C, 60–70% relative humidity and L16: D8 light cycle. Plants were grown in a glasshouse to the 7–9 leaves stage and used for all experiments. Green peach aphid was reared on cabbage in cages made from plastic and covered by anti-insect white mesh with external dimensions of 40 cm × 40 cm × 40 cm.

### 4.2. Insect Culture

*Myzus persicae* for experiments were obtained from the Department of Primary Industries and Regional Development, Entomology Branch (Western Australia) and maintained on potted cabbage seedlings in a greenhouse that were placed into large cages (210 cm × 90 cm) covered by anti-aphid mesh and provided with a control light system set at L16: 8D photoperiod, at the glasshouse temperature 23–25 °C, located at Murdoch University (Western Australia).

*Aphidus colemani* (Hymenoptera, Braconidae) and *Aphelinus abdominalis* (Hymenoptera, Aphelinidae) were commercially obtained from Biological Services (South Australia) as mummies and maintained on potted cabbage plants infested with *M. persicae* as hosts. Mummies of wasps were removed from the plant leaves on the 12th day for the *A. colemani* and 15th day for the *A. abdominalis* of the parasitism, and placed in open 9 cm Petri dishes inside a small cage of 40 cm × 40 cm × 40 cm, in greenhouse conditions (23–25 °C, 60–70% RH, 16:8 L:D) until emergence. Then, the parasitoids were allowed to mate in the cage for one day with provided 50% honey solution for feeding. After that, the parasitoid was held individually in glass vials (one wasp per vial), a small piece of cotton attached to the vial cap for the drop of 50% honey solution to feed the parasitoid until tested. Female wasps were used for the Y-shape olfactometer choice test [9].

### 4.3. Volatiles Collection and GCMS Analysis using HS-SPME

#### 4.3.1. VOCs Extraction with HS-SPME

The analysis of volatiles was focused on cabbage for infested and uninfested plants with the green peach aphid. Cabbages were placed individually into 4 L glass jars, and one plant in each jar was analyzed. For each glass jar, a 5 mm port was drilled into the side, into which a septa (20633 Thermogreen^®^ LB-2 Septa, plug) was placed and used for the collection of infested and uninfested plant VOCs. Aluminum foil of 100 m × 44 cm (Vital Packaging Company) was used to carefully cover and wrap the surface of the top of the plant pot, and the glass jar placed upside down on the plant. The reason for selecting glass jars is that it is easy to capture the VOCs emitted and also easy to wash, clean and oven-dry them at 100 °C for a minimum of 30 min to sterilize. VOCs were extracted from samples, which were infested and uninfested cabbage plants with *M. persicae*. For extracting VOCs from samples, headspace technique analyses were used with three replicates in all experiments, for profiling and characterization of VOCs from both plants. The identification of VOCs was conducted with the SPME fiber by extracting the compound from the headspace of treatments. Three phase fibers 50/30 μm divinylbenzene/carboxen/polydimethyl siloxane (PDMS/CAR/DVB; Sigma-Aldrich, Australia, catalogue number 57347-U) coating was selected for volatiles released from infested and uninfested plants. The SPME fiber is commonly used and this three phase fiber was selected because it was being used for the analysis of a wide range of analysts. The fibers were first conditioned at the range of operating temperature recommended by the manufacturer, before analyses were conducted. For optimizing various conditions, the sealing time was optimized to 2.30 h under laboratory temperature 25 ± 1 °C, and the SPME fiber was exposed to the headspace of the samples by inserting the SPME into the jar through the septum for two hours to extract the VOCs, which characterized the optimum extraction time. The desorption time of SPME fiber was 5 min in the GC injection port. The SPME was used because it is a fast, simple and modern tool for GC-MS analysis.

#### 4.3.2. Samples Analysis with GC-MS

The analysis of VOCs obtained by HS-SPME was performed on a gas chromatography mass spectrometer (GC Agilent GCMS 7820A) equipped with MS detector 5977E (Agilent Technologies, USA) and a DB-35ms column (30 m × 250 μm × 0.25 μm) (Santa Clara, CA, USA). The fiber was desorbed in the splitless injector 270 °C of GCMS with other operation conditions. The initial temperature of the column was 50 °C and held for 2 min, then increased to 250 °C at 5 °C min^−1^ and held for 5 min at 250 °C. Helium gas (He) was used as a carrier and supplied by (BOC Gas, Sydney, Australia) and the flow rate of the column was 1:1 mL/min, while the splitless was 20 mL/min at 1.5 min and the total GC-MS run time was 45 min. The calibration of the SPME fiber was performed by injecting the *n*-alkanes standard C7–C30.

HS-SPME/GC-MS analysis of the VOCs were identified by using AMDIS software version 2.72 and the US National Institute of Standards and Technology (NIST) 2014 MS database. The VOCs were confirmed by comparing GC retention time data with those of authentic standards or from the published literature [44].

### 4.4. Evaluation of Olfactory Responses of M. persicae and Its Parasitoids

A glass Y-tube olfactometer was used to determine the responses of *M. persicae* and its two species parasitoids, *A. colemani* and *A. abdominalis*, to each of the following pairs of plant treatments. For the aphid responses, the test was (1) infested (cabbage plants infested with *M. persicae*) versus clean (filter) air; (2) non-infested versus clean air; and (3) infested versus non-infested plants (Figure 5). For the test of parasitoid wasps, *A. colemani* and *A. abdominalis*, (1) infested plant versus clean air; (2) non-infested plant versus clean air; and (3) infested versus non-infested plants. Bioassays were used to compare their olfactory responses to VOCs released from uninfested plants versus clean air or infested plants with *M. persicae* versus uninfested plants. The infested cabbage plants that were used in this study contained aphids.

Volatile preference experiments were made using a glass Y-tube olfactometer as previously described [45], with a 7 cm arm length and 2 cm internal diameter, ground glass fitting for the air that passed 200 mL/min through each arm, controlled by air flow meter (SCFH AIR, Dwyer Instruments, Michigan City, IN 46360, USA) (Figure 5). Each arm tube was connected to a glass chamber (2 L desiccator). Couples of blend VOCs (released from different plant treatments) were presented in a sealed glass chamber (2 L each) at the end of either arm. The compressed air was filtered by using activated charcoal passed through two glass chambers, before the treatment plant could be introduced, and then the air passed through the olfactometer. After assembly, the olfactometer was left to stabilize for 15 min prior to use [46].

The Y-tube olfactometer work was carried out under the same conditions as the glasshouse conditions. The area surrounding the olfactometer (below and around) was covered by white paper and white light was placed directly over the olfactometer. For the bioassay, a single aphid or single parasitoid was introduced into the main arm of olfactometer and pushed 1–2 cm inside the main arm. Each aphid or wasp was given up to 3 min in the olfactometer to respond. Once an individual moved beyond 2 cm and into one of the Y-tube arms, it was considered to have made a choice for the conforming plant treatment in that arm. Non-responders that did not make a choice in 5 min were discarded and excluded from the statistical analysis (non-responsive parasitoids counted in statistical analysis in the experiment of comparison of clean air with the uninfested plant).

Three replicates and 30 adults of wingless aphid *M. persicae* were assayed for each replicate, and each aphid was tested only one time. Every 10 aphids were assayed, the volatile treatment resources were removed, and all glass vessels cleaned with ethanol, then washed with water and oven dried at 100 °C for a minimum of 30 min. For the comparison, three replicates were carried out on different days using new aphids and fresh infested and non-infested plants. All plant resources were the same age and same size.

The same procedure above was carried out for the parasitoid *A. colemani* and *A. abdominalis*. Additionally, three replicates were used for the parasitoids with 15 wasps for each replicate and wasps were used only once. Throughout the experiments, after all 15 wasps were assayed for each replicate, the apparatus was cleaned with water and ethanol, then dried and heated in the oven at 100 °C for more than 30 min. Statistical significance between wasp responses to pairwise combinations of plant treatments was determined using Chi-square tests at the 5% level.

### 4.5. Statistical Analysis

To identify the differences in the emission of volatile compounds between uninfested and infested cabbage by green peach aphids, all peak area analyses were performed with MetaboAnalyst software for the *p*-value, principal component analysis (PCA and PLS-DA) and the hierarchical clustering heat map [47]. The differences in the results were compared by using the least significant difference test (*p* ≤ 0.05) for determining the means between infested and uninfested plants. The peak area was divided by 100,000 for every single compound that obtained from GC-MS and subjected to analysis of variance (ANOVA) using Genstat software version 10 (VSNI International Limited, UK) and the least significant difference (LSD) was used at 5% probability level. The data of the Y-tube olfactometer bioassays were analyzed for preference (aphid *M. persicae* and their parasitoids *A. colemani* and *A. abdominalis* choice between two treatments tested) using the Chi-square goodness of fit test by using SPSS software version 24.0.

## 5. Conclusions

The HS-SPME with GC-MS analysis for the volatiles described the differences between the infested and uninfested cabbage plants and their role in attracting natural enemies of aphids. Collection of volatiles from cabbage occurred by using HS-SPME to detect volatiles compounds between uninfested and plants infested with *M. persicae* and examined the attraction of natural enemies. A total of 28 VOCs were identified in cabbage plant treatments, by using HS-SPME combined with GC-MS. The parasitoids *A. colemani* and *A. abdominalis* laid eggs within the body of *M. persicae* and immature stages completed development inside the hosts, eventually killing them by feeding the wasp larva inside the aphids; the parasitoid pupates inside the aphid mummy and they emerges as an adult. To detect and locate hosts, it is believed that *A. colemani* and *A. abdominalis*, as with many parasitoids, rely on odors released from infested plants as a response to aphids feeding. The results indicated that the preferences of *A. colemani* and *A. abdominalis* to infested plants with *M. persicae* compared with uninfested plants and clean air by using an olfactometer. The results showed that parasitoids can discriminate the infested cabbage and significantly respond to the plant odor. Thus, we believe that aphid parasitoids can find damaged plants and then detect aphids on the plant-by-plant odor. It is likely that the natural enemies’ search for aphid infestation may start before landing on the uninfested plant, because parasitoids will first find a damaged plant and then begin searching for aphids. For this reason, many aphid parasitoids efficiently search for damaged plants where aphids will be present, as explained by [20].

## Figures and Tables

**Figure 1 molecules-27-01522-f001:**
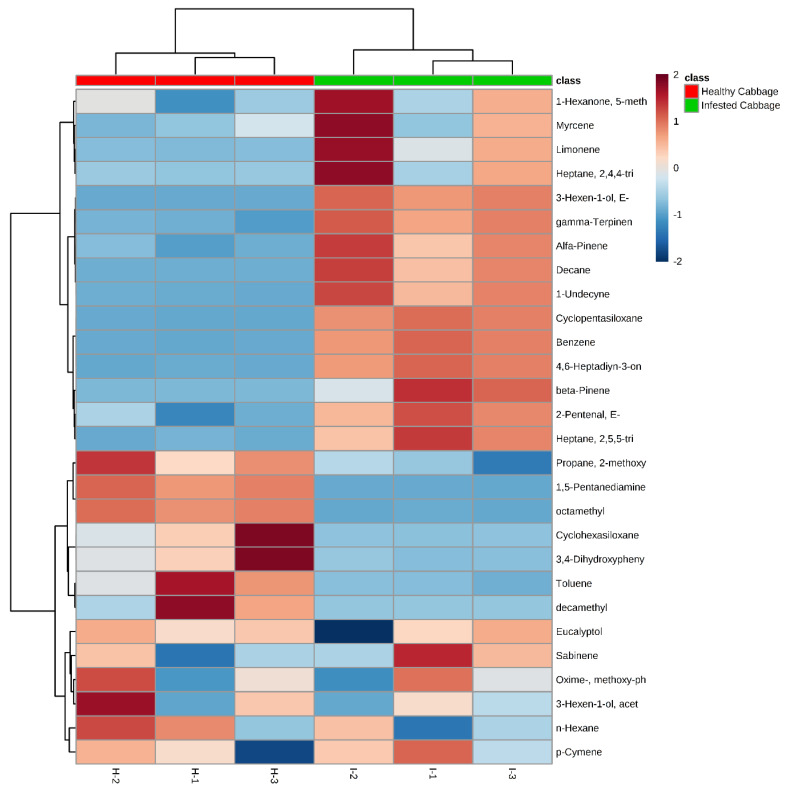
Clustering result is shown as a heat map of volatile compounds released from uninfested and infested cabbage with green peach aphid *M. persicae*. Each volatile compound’s peak area detected by GC-MS is shown by colors.

**Figure 2 molecules-27-01522-f002:**
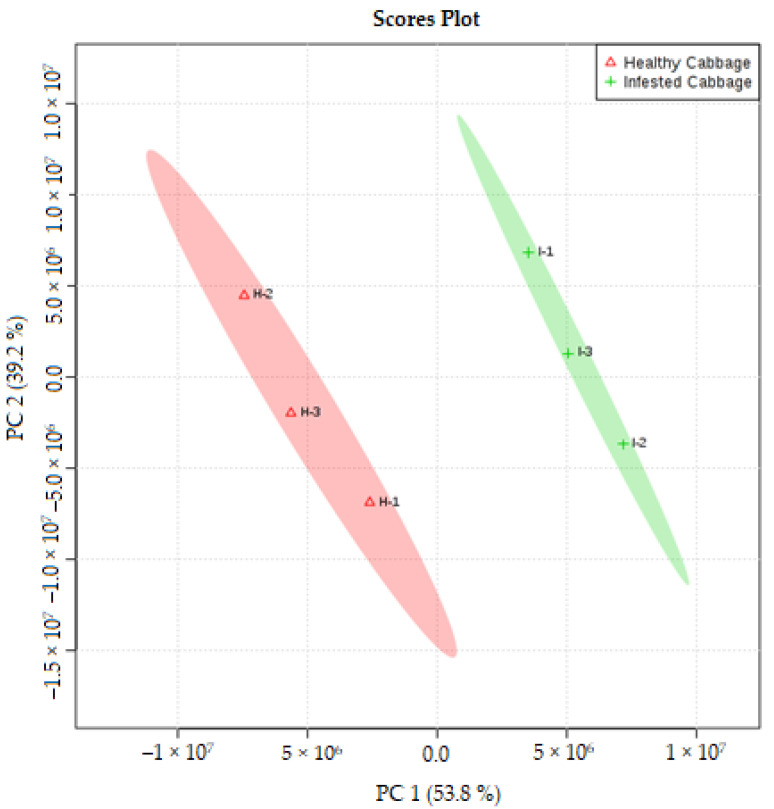
Principal component analysis (PCA) scatter plots reveals between the volatile compounds detected in uninfested and infested cabbage with *M. persicae*. PCA was applied to VOCs from three replicates uninfested and three replicates infested cabbage plants. Red and green circles show results of K-means clustering with k = 2 clusters.

**Figure 3 molecules-27-01522-f003:**
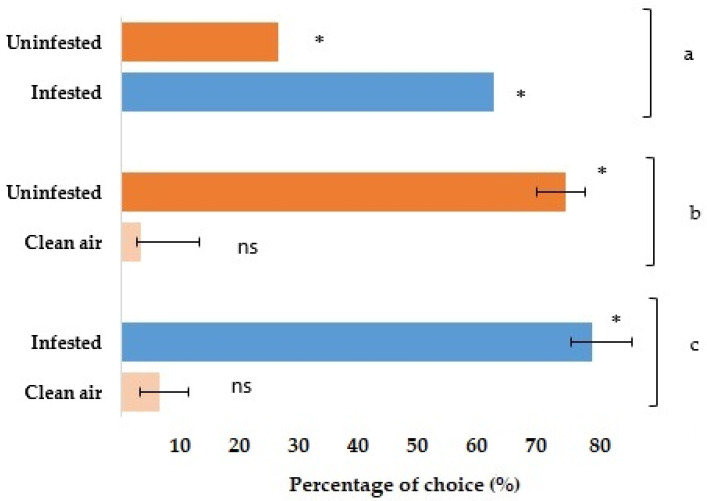
Olfactory response of green peach aphid *M. persicae* in Y-tube olfactometer experiments to volatiles released from infested and uninfested cabbage (**a**) uninfested versus infested plants (**b**) uninfested versus clean air (**c**) infested versus clean air. All treatments presented with standard deviation (SD) bar. Asterisks (*) indicates significant difference *p* < 0.05 (Chi-square test).

**Figure 4 molecules-27-01522-f004:**
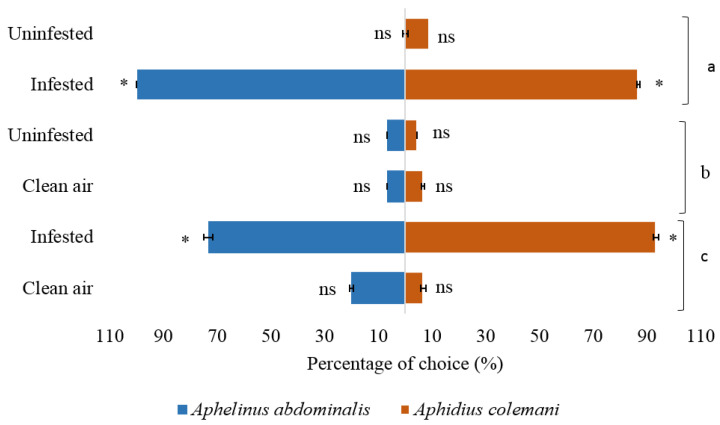
Olfactory response of two parasitoids *Aphidius colemani* and *Aphelinus abdominalis* in Y-tube olfactometer experiments to volatiles released from infested and uninfested cabbage *B. oleracea* (**a**) uninfested versus infested plants (**b**) uninfested versus clean air (**c**) infested versus clean air. All treatments presented with standard deviation (SD) bar. * indicates significant difference *p* < 0.05 (Chi-square test).

**Figure 5 molecules-27-01522-f005:**
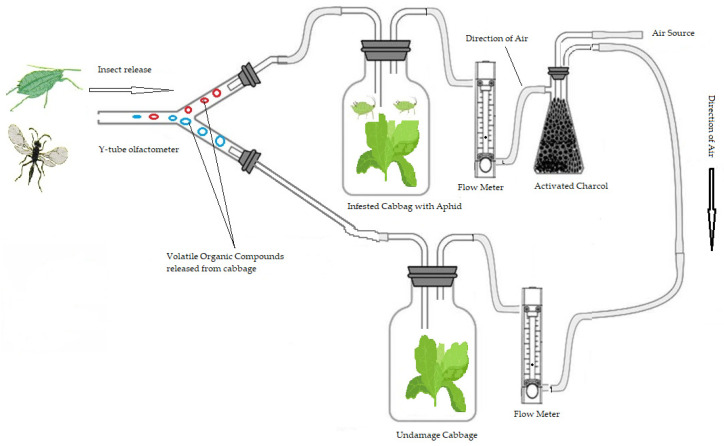
The diagram of the olfactometer, including the glass Y-tube where the aphid *Myzus persicae* and the parasitoid *Aphidus colemani* and *Aphelinus abdominalis* were released individually and exposed to two plant VOCs, blends from uninfested and plants infested with *M. persicae* as shown by the blue and red small circles.

**Table 1 molecules-27-01522-t001:** Volatile compounds detected in the headspace of infested and uninfested cabbage with *M. persicae* by using solid phase microextraction (SPME).

No	Compound Name	RT ^1^	Uninfested Plant Area ± SD ^2^	Infested Plant Area ± SD	LSD ^4^	*p*-Value
1	Propane, 2-methoxy	3.12	23.10 ± 3.13	7.84 ± 2.70	11.45	0.020 *
2	*n*-Hexane	3.28	15.38 ± 4.21	8.40 ± 3.83	15.8	0.199
3	Benzene	3.61	72.20 ± 1.55	601.75 ± 28.09	78	0.305
4	3-Hexen-1-ol, (E)	6.38	ND ^3^	28.83 ± 1.51	4.197	0.223
5	4,6-Heptadiyn-3-one	9.33	90.28 ± 2.26	601.75 ± 28.09	78.2	1.211
6	Toluene	11.02	12.50 ± 3.48	1.65 ± 0.31	9.7	0.653
7	Oxime-, methoxy-phenyl	12.43	757.69 ± 322.83	680.68 ± 300.96	1223.9	0.200
8	2-Pentenal, (E)-	12.49	16.86 ± 0.82	23.36 ± 0.76	3.105	0.136
9	Alpha-Pinene	13.32	24.44 ± 4.96	131.41 ± 16.53	47.87	0.003 *
10	Sabinene	13.47	72.54 ± 34.72	137.59 ± 37.07	140.8	0.377
11	Myrcene	15.22	20.15 ± 7.96	68.45 ± 30.99	88.7	0.046 *
12	beta-Pinene	16.25	ND	55.75 ± 17.03	47.24	0.930
13	1-Hexanone, 5-methyl-1-phenyl	16.81	21.05 ± 3.78	35.38 ± 7.44	23.14	0.004 *
14	*p*-Cymene	17.28	422.85 ± 144.03	564.67 ± 82.08	459.7	0.339
15	3-Hexen-1-ol, acetate, (Z)	17.48	394.93 ± 152.39	245.99 ± 62.11	456.3	0.277
16	Eucalyptol	19.97	129.50 ± 5.22	96.14 ± 34.98	98.1	0.036 *
17	Limonene	20.38	14.66 ± 1.92	247.26 ± 84.09	233.2	0.003 *
18	Decane	23.57	ND	39.31 ± 5.50	15.25	0.020 *
19	gamma-Terpinen	24.81	9.03 ± 1.70	56.55 ± 3.68	11.23	0.007 *
20	Heptane, 2,4,4-trimethyl	26.24	3.75 ± 1.44	91.50 ± 45.46	126.1	0.001 *
21	Cyclopentasiloxane, decamethyl	27.84	1.95 ± 0.23	314.91 ± 12.00	33.29	0.212
22	1-Undecyne	30.22	2.68 ± 0.52	110.55 ± 13.59	37.72	0.036 *
23	Heptane, 2,5,5-trimethyl	30.82	2.17 ± 0.43	33.82 ± 4.85	13.5	0.630
24	Cyclohexasiloxane	34.24	123.62 ± 53.60	1.16 ± 0.17	148.6	0.301
25	3,4-Dihydroxyphenylglycol	37.29	20.15 ± 7.96	1.72 ± 0.41	22.09	0.286
26	1,5-Pentanediamine	40.10	249.45 ± 12.70	10.33 ± 0.64	35.27	0.127
27	octamethyl	42.66	565.00 ± 22.07	7.89 ± 2.42	61.6	0.129
28	decamethyl	41.43	113.05 ± 55.42	ND	153.7	0.401

^1^ RT indicated to the retention time of compounds. ^2^ SD referred to the standard deviation of peak area calculated from three replicates. ^3^ ND referred to not detected. ^4^ LDS referred to Least Significant Difference at 0.05 level.* indicated to the significant different 5%.

## Data Availability

All data are contained within the article.

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
