# Peer review of "Response of Aphid Parasitoids to Volatile Organic Compounds from Undamaged and Infested *Brassica oleracea* with *Myzus persicae"

_molecules, 2022, doi:10.3390/molecules27051522_

Round 1

Reviewer 1 Report

This study determines the volatile compounds produced by aphid-infested and uninfested cabbage plants and investigates the attraction of aphid and parasitoids to aphid-infested and uninfested cabbage plants in an olfactometer. The idea and the findings of this study is not new; however, it is still an important study. The authors demonstrated that aphids and parasitoids prefers volatiles from by aphid-infested cabbage. It would have been interesting if the authors had elucidated the volatiles produced by aphid-infested cabbage (with aphids) and aphid-infested cabbage (without aphids) as well as their attraction of aphids and parasitoids. It is possible that some of the volatiles detected could have been produced by the infesting aphids.  Some of the sentences need to be revised for clarity. Also, the heat map adds little value to the manuscript. Nevertheless, the manuscript can be accepted after revision of the texts. Specific comments are mentioned below.   

 Line 61-63: Revise sentence, “Previous studies have concluded that natural enemies can recognize the VOCs that are released from infested plants and confirmed by using the olfactometer…”

Line 67 Replace bring with lure in sentence

Line 71: Delete the word “both’ from the sentence

Line 72: Revise to, “The parasitoid wasps…” Also, insert space between genus and species name  A.abdominalis

Line 77: Reed [8]…

Lines 87-91: The objectives are not well-written. One can barely understand what the authors intended to say.

My suggestion, “Therefore, the study aimed to determine the VOCs released from the cabbage plants infested by M. persicae and the uninfested plants and to elucidate the responses of M. persicae and their parasitoids (A. colemani and A. abdominalis) to aphid-infested and uninfested cruciferous plants in a Y-tube olfactometer.”

What do the authors mean by cruciferous plants? Cabbage? It is important to be specific.  

Results:

Line 107: There was a significantly larger quantity…” How significant? Please show statistics

Figures 1 and 2 do not add value to the results and the manuscript. I could not see the value of the heat map.

Infested and uninfested should be used in the manuscript instead of infested and healthy

Line 147-149: Revise the sentence to “These results indicated that green peach aphids in cabbage were significantly (Chi-Square (2) =18.61, df= 1 and P<0.0005) more attracted to the VOCs released from infested plant (80%) rather than clean air (7%).”

Lines 163-164: Delete this sentence.

Line 168: Insert respectively “…7% and 20%, respectively,…”

Lines 163-185: It is possible that the aphid volatiles were also contributing to the attraction of the parasitoids. The authors should investigate this. Did the authors remove all infesting aphids from plant materials before carrying out the olfactory study? Please clarify in the materials and methods section

Discussion

It might be interesting to discuss whether some of the volatile compounds detected in this study are known attractants to aphids and the parasitoids.    

Materials and methods

The materials and methods are appropriate and written in detailed.

Reviewer 2 Report

The manuscript entitled -Response of Aphid Parasitoids to Volatile Organic Compounds from Undamaged and Infested Brassica oleracea with Myzus persicae- by Qasim Ahmed and coworkers, is about the effect of some molecules on the preference of an aphid and two parasitoids.

The manuscript is quite interesting with some novelty, but I must admit that sometimes is difficult to read and to understand idea that guide the authors, so my suggestion is to reconsider the manuscript after a major revision.

Some points to be considered:

―  Why the name of the chemical molecules is written with capital letters, and sometimes two different names correspond to the same molecules? See line 107 and entry 4 in table 1, I strongly advise to follow the IUPAC nomenclature rules.

― Lines 101-104. Some molecules decrease the amount when infested, see table 1.

― Table 1. Add the number of repetitions used to calculate the STD. P-value, why in some case is missing entry 5, 10, 14, 25, and why in some case is present even if some data are missing entry 4, 12, 19, and what about entry 29? The LSD (line 369) require that the null hypothesis must be rejected, but I could find such statistical analysis.

― Lines 116-124. Here is reported the correlation among the molecules for the unifested and infested cabbage. I don’t understand the reason to use each single measurement and not use the mean calculated in table 1.  In addition, the report of the results obtained from this type of analysis is missing. How I have to use the results obtained?

― Lines 124-126. The same for the PCA analysis, is not clear the results obtained where are commented.

― Figures 1 and 2 are in the manuscript but they don't seem so important as results and in the discussion, I suggest rethinking the meaning of these figures.

― Lines 143-146. Add the number of replicates.

― Figure 3. The SD it seems not centered on the mean value. The horizontal axe lack of units

― Lines 213-215. This sentence is not clear to me.
